# Study of Construction of Innovative Barite/Waterborne Polyurethane/Low-Density Polyethylene Composites for Enhanced X-Ray Shielding Performance

**DOI:** 10.3390/polym17040451

**Published:** 2025-02-08

**Authors:** Xi Xu, Shujin Shi, Xianrong Yang, Huan Shuai, Gaoxiang Du, Jiao Wang

**Affiliations:** 1School of Materials Science and Technology, China University of Geosciences, Beijing 100083, China; 2103220017@email.cugb.edu.cn (X.X.);; 2Beijing Yiyi Star Technology Co., Ltd., Beijing 100089, China; 3School of Basic Education, Beijing Polytechnic College, Beijing 100042, China

**Keywords:** waterborne polyurethane, modification, barite, compatibility, X-ray shielding

## Abstract

X-rays’ high-energy nature poses risks to human health. Traditional X-ray shielding materials often contain toxic lead and have drawbacks like bulkiness and rigidity. Consequently, there is an increasing need to develop lightweight, non-toxic, flexible, and efficient shielding materials. In this study, we modified barite with waterborne polyurethane (WPU) and systematically investigated the effects of WPU on barite’s properties. The modification with WPU not only reduced the tendency of barite (B) to agglomerate but also enhanced its compatibility with polymers, thereby significantly improving the mechanical properties of LDPE/WPU-B composites. Compared to unmodified barite in LDPE/B composites, the tensile and flexural modulus of the LDPE/WPU-B composites increased by 22.31% and 29.64%, respectively. With 20% WPU-modified barite, the radiation shielding efficiency increased by 5%. When the WPU-B content reached 40%, the shielding efficiency of the LDPE/WPU-B composite exceeded 90% for tube voltages ranging from 60 kV to 120 kV, achieving a lead equivalent of 0.38 mmPb at 100 kV. This novel LDPE/WPU-B composite has great potential for low-dose radiation shielding applications.

## 1. Introduction

High-energy X-ray radiation plays a crucial role in various fields such as disease diagnosis, treatment, and scientific research [1,2,3,4]. While X-ray radiation brings benefits to humanity, it also poses serious threats to human health and the environment. Due to the extremely high energy levels, X-rays can easily penetrate the human body, potentially causing tissue and organ damage, leading to radiation injuries and even cancer [5]. Lead, with its high atomic number and non-radioactive properties, is widely used in radiation shielding applications. According to radiation protection theory, materials with high density (ρ) or elements with a high atomic number (high-Z elements) can more effectively attenuate and absorb photon energy through mechanisms such as the photoelectric effect, Compton scattering, and electron pair production [6]. However, lead’s heavy weight and toxic nature increase discomfort for users and pose a risk of lead poisoning [7,8]. Therefore, developing lightweight, non-toxic, elastic, and effective X-ray shielding materials is of utmost importance [9].

Polymer composites are attracting increasing attention due to their advantages of being lightweight and elastic. Researchers are investigating polymer-based composites incorporating high-Z element compounds as potential replacements for Pb in effective X-ray attenuation applications [9,10,11]. Among various fillers, barite (BaSO_4_) has the characteristics of a heavy barium nucleus, a high number of atoms per unit volume, and a high probability of photoelectric and Compton effects and has therefore attracted much attention. Lopresti et al. [12] enhanced compatibility with resins and improved dispersion in epoxy resins using stearic acid and sodium lauryl sulfate coating on barite. Under acceleration voltages of 80–120 kV, these composites performed better than steel; then, researchers in [13] coated stearic acid and sodium dodecyl sulfate on the surface of barite by using the liquid-assisted grinding (LAG) method, significantly enhancing the compatibility and dispersion of these functional additives in the resin matrix. Qu et al. [14] developed a lead-free X-ray protective material containing 25 wt% BaSO_4_, achieving an equivalent to approximately 0.1 mmPb of lead. Pulford et al. [15] coated textile substrates with composites containing 70 wt% BaSO_4_, achieving about 50% X-ray attenuation at a 70 kV peak voltage. Kinnune et al. [16] prepared low-dose radiation shielding plastic materials from barite tailings, combining ecological and economic benefits for the safe storage of low-level radioactive waste. Messele et al. [17] embedded bismuth oxide (Bi_2_O_3_) and BaSO_4_ metal particles into polychloroprene (CR) to obtain a dual-filler composite with X-ray radiation shielding performance. This material showed synergistic effects at 20 parts particle loading (10 Ba/10 Bi-CR), with enhanced attenuation across nearly all tube voltages, and the sample density was 7–8 times lower than that of commercial lead-based shielding. Nakamura et al. [18] manufactured a dual-layer radiation shielding material with BaSO_4_ and tungsten, which was about 13% lighter than a 0.5 mm thick pure lead plate while providing equivalent shielding efficiency. Mungpayaban et al. [19] synthesized X-ray shielding materials by incorporating BaSO_4_/amorphous cellulose (Ba/AC) into natural rubber, effectively absorbing most incident X-ray beams due to the well-dispersed and reduced aggregation of BaSO_4_ particles. For the study of waterborne polyurethane, Yu et al. [10] developed a PDMS/BTO-coated polyester fabric with a 1.1 mm thickness and a sandwich structure. This fabric demonstrated X-ray shielding performance comparable to that of 0.35 mm of lead at both 80 kV and 100 kV. Additionally, it was 42% lighter compared to a 0.35 mm lead sheet.

This paper aims to improve the dispersion of barite (B) in polymers and enhance radiation shielding effects by using a chemical coating modification method with aqueous polyurethane (WPU) for modification, and then analyzes and discusses the coating mechanism. This sets our research apart from others; additionally, the modified ultrafine barite powder was mixed with low-density polyethylene (LDPE) and processed using open mixing and compression molding to prepare LDPE/WPU-B materials, with systematic studies conducted on their mechanical and radiation shielding properties.

## 2. Materials and Methods

### 2.1. Experimental Materials

The raw materials utilized in this study include ultrafine barite powder supplied by Hubei Gucheng Xinhe Co., Ltd. (Xiangyang, China), waterborne polyurethane provided by Anhui Anda Huatai New Material Co., Ltd. (Hefei, China), kerosene furnished by Jingtu Hengsheng Trading Co., Ltd. (Beijing, China), and low-density polyethylene (LDPE) sourced from SINOPEC Beijing Yanshan Company (Beijing, China).

### 2.2. Experimental Procedure

Preparation of WPU-B powder. A wet chemical coating method was used for the graft modification of ultrafine barite powder (as shown in Figure 1). The ultrafine barite powder and distilled water were added to a beaker, stirred evenly, and then placed in a water bath at 90 °C for stirring. After uniform stirring, a certain amount of aqueous polyurethane was added and stirred continuously for 1 h. The mixture was then cooled to room temperature, filtered, dried at 90 °C, and dispersed.

Preparation of LDPE/B and LDPE/WPU-B composites. The process of preparing the composites included mixing, open milling, and compression molding. A certain amount of dried WPU-B and LDPE was mixed together and placed in an open mill for processing. For barite (B), the addition was 20 wt%, while for WPU-B, the added amounts were 20 wt%, 40 wt%, and 60 wt%. After milling, the samples were taken out and subjected to compression molding to obtain LDPE/B and LDPE/WPU-B composites.

The abbreviations of samples mentioned in this article and their corresponding explanations are shown in Table 1.

### 2.3. Testing and Analysis Methods

The dispersibility of the modified barite particles in the organic phase can be characterized by their settling time in kerosene. The settling time referred to herein denotes the duration required for a specified quantity of dispersed powder particles, once added to kerosene and thoroughly mixed, to sediment to a predetermined level when left undisturbed. The longer the settling time, the better the dispersion of the modified barite in the organic phase. A certain amount of B and WPU-B particles was added to a beaker containing 50 mL of kerosene and magnetically stirred for 10 min, and then 25 mL of the suspension was transferred into a 25 mL graduated cylinder. The timer started, and when the 5 mL mark became clearly visible, the timer was stopped. This recorded time is the settling time.

The changes in particle size before and after WPU modification were tested using a Bettersize laser particle size analyzer.

The contact angles were measured using a Kruss (Hamburg, Germany) DSA100 contact angle measuring instrument. The fitting method used was Ellipse.

Infrared Measurement: Infrared spectroscopy of barite before and after WPU graft modification was performed using a Thermo Fisher (Waltham, MA, USA) Nicolet iS20 spectrometer, with a wavenumber range of 400 to 4000 cm^−1^.

The crystal structures of the powder before and after WPU modification were characterized using an Ultima IV X-ray diffractometer (Rigaku, Tokyo, Japan), with a scanning speed of 10°/min over a range of 10° to 90°.

The microscopic morphology of the samples was observed using a Hitachi JSM-7610F scanning electron microscope (Hitachi Ltd., Tokyo, Japan). Sample preparation methods were as follows: (1) Powder: Modified barite powder was added to anhydrous ethanol to form a suspension, which was then ultrasonically treated and dripped onto a silicon wafer using a pipette, followed by gold sputtering (20 mA, 120 s). (2) Specimens: Molded specimens were soaked in liquid nitrogen for 5 min, removed, and subjected to brittle fracture. The fractured surfaces were then prepared for observation, followed by gold sputtering (20 mA, 120 s).

The thermogravimetric curve of the powder was tested. Thermogravimetric curves of B, the WPU-B powder, and the specimens were analyzed using a NETZSCH (Selb, Germany) TG 209 F3 instrument, with a heating rate of 20 °C/min and a temperature range of 30 °C to 1000 °C in a N_2_ atmosphere.

The mechanical properties of the composite materials were tested. Tensile and flexural properties of LDPE/B and LDPE/WPU-B composites were tested using a universal testing machine by WANCE (Shenzhen, China), with a tensile rate of 10 mm/min and a flexural rate of 5 mm/min. The dimensions of the test specimens are illustrated in Figure 2.

The radiation shielding performance of the composite material was tested. The energy attenuation behavior of X-ray photons passing through shielding materials (attenuators) follows the Beer–Lambert law, expressed as follows [20]:$$I=I_{0}\cdot e^{-\mu }$$
where *I* represents the transmitted radiation dose rate, *I*_0_ is the incident radiation dose rate, and *μ* is the linear attenuation coefficient of the shielding material in cm^−1^. Based on the Beer–Lambert law, the radiation resistance testing method in this study was conducted according to the “YY/T 0292.1-2020 Medical Diagnostic X-ray Radiation Protection Equipment Part 1: Determination of Material Attenuation Performance” standard [21]. An MG325 X-ray machine was used as the emission source, and a TW34069-2.5 ion chamber dosimeter was used to measure the attenuation effect by evaluating the change in the radiation dose rate through sample measurements. Figure 3 below shows the narrow beam conditions and test sample sizes (as shown in Figure 3a) for radiation testing. As shown in the Figure 3a, a represents the distance from the test object to the reference point of the radiation detector on the beam axis, d denotes the diameter of the radiation detector, and t indicates the diameter of the radiation beam on the far end surface of the test object. During the test, the incident radiation dose rate *I*_0_ without samples was measured first. Then, the sample (as shown in Figure 3b) was placed at position 4, and the transmitted radiation dose rate *I* was measured. The measurement distance during the test was 2 m, and the beam current was set at 15 mA. When the tube voltage was set to 60 kV, 80 kV, 100 kV, and 120 kV, respectively, the incident radiation dose rate *I*_0_ without shielding was approximately 0.167 mGy/s, 0.294 mGy/s, 0.447 mGy/s, and 0.617 mGy/s.

Various indicators for X-ray protection, such as attenuation efficiency, linear attenuation coefficient, mass attenuation coefficient, and half-value layer, can be calculated based on the measured dose rate using the following formulas [20].
(1)Attenuation efficiency (RPE): Attenuation efficiency refers to the ratio of the attenuated X-ray dose rate (*I*) after a certain dose of X-ray passes through the sample to the incident X-ray dose rate (*I*_0_). The unit is %, and the calculation formula is as follows:
$$RPE=\frac{I_{0}-I}{I_{0}}\times 100\%$$
(2)Linear attenuation coefficient (LAC, μ): The linear attenuation coefficient is used to describe the X-ray attenuation effect per unit thickness of material and is calculated using the following formula:
$$\mu =\frac{-\ln\frac{I}{I_{0}}}{d}\times 100\%$$
where (*d*) is the sample thickness in cm.
(3)Mass attenuation coefficient (MAC, *μ_m_*): The percentage of reduction in X-ray intensity per gram per centimeter thick of the absorbing substance is the mass attenuation coefficient, with the unit of cm^2^‧g^−1^. The calculation formula is as follows:
$$\mu_{m}=\frac{\mu }{\rho }\times 100\%$$
where (*ρ*) is the sample density in g·cm^−3^.

(4)Half-Value Layer (HVL): The half-value layer is the thickness of the material required to reduce the X-ray intensity by half, expressed in cm, calculated as follows:


$$HVL=\frac{\ln2}{\mu }$$


(5)Lead Equivalent: The lead equivalent is the thickness of lead that provides identical shielding effectiveness to that of the test material when both are exposed to the same X-ray source, expressed in mmPb. The calculation method of lead equivalent involves irradiating the material to be tested with X-rays, measuring its radiation dose, and comparing it to the radiation dose of a lead plate of known thickness under the same conditions, thereby calculating the lead equivalent of the material to be tested.

## 3. Results and Discussion

### 3.1. Discussion of Modification Effect

The quantity of the modifier significantly influences the effectiveness of surface modification. An insufficient modifier quantity hinders the formation of a complete adsorption layer on barite particles, thereby failing to achieve optimal surface modification. Conversely, an excessive amount of modifier can also diminish the modification effect. This study systematically examines the impact of varying amounts of waterborne polyurethane (WPU) modifier on barite particle modification. As illustrated in Figure 4a, the particle size distribution (PSD) exhibits an increasing trend with higher WPU concentrations. Specifically, as the WPU concentration increases from 1.5 wt% to 3.0 wt%, the particle size distribution broadens and the average particle size increases due to the formation of a polyurethane adsorption layer on the barite surface. Figure 4b demonstrates that the settling time of barite particles in kerosene increases with WPU addition, peaking at 105 s when the WPU concentration is 1.5 wt%. Beyond this point, further increases in WPU concentration lead to a decrease in settling time. The static contact angle measurements, conducted using a contact angle measuring instrument (Figure 4b), reveal that unmodified ultrafine barite powder exhibits hydrophilic properties with contact angles of 29° and 29.1°. Upon adding the WPU modifier, the contact angle increases, indicating improved dispersion of ultrafine barite particles. At a WPU concentration of 1.5 wt%, the contact angles reach maximum values of 66° and 68°. However, when the amount of aqueous polyurethane (WPU) exceeds 1.5 wt%, the contact angle of ultrafine barite powder decreases. At a WPU addition level of 1.5 wt%, the grafting of WPU on the barite (B) surface reaches saturation, forming a complete polyurethane film. Further increases in WPU concentration result in interactions between the WPU layers on adjacent WPU-B particles, leading to particle aggregation and adhesion. This phenomenon is evidenced by an increase in particle size and a reduction in settling time. Moreover, the increased gaps and surface irregularities between aggregated particles contribute to the observed decrease in contact angle [22,23].

### 3.2. Analysis of Modification Mechanism

To explore whether aqueous polyurethane modification changes the crystal structure of barite, we performed XRD testing on ultrafine barite powder before and after modification. The results are presented in Figure 5. According to the XRD results, the diffraction peaks of unmodified barite at positions 2θ= 25.86°, 26.86°, 28.78°, 42.59°, and 42.91° align precisely with those of the barite card PDF #98-000-0106, corresponding to the crystal faces (210), (102), (211), (113), and (122) of barite. The diffraction peaks of barite modified with 1.5 wt% aqueous polyurethane and unmodified barite align closely, indicating that the addition of aqueous polyurethane does not alter the crystal structure of barite.

In order to explore the influence of waterborne polyurethane on the dispersibility of barite powder, we also conducted SEM tests on the powder before and after modification. The SEM results of the test shown in Figure 6a,b are the morphologies of unmodified barite. It can be seen from the figure that before modification, the barite particles agglomerated with each other, and there were many agglomerates larger than 1 μm. The SEM images after modification by polyurethane (Figure 6c,d) show that the agglomerated particles of barite were opened, and the particles were evenly dispersed, with only a small amount of barite agglomerated particles close to 1 μm. SEM indicates that waterborne polyurethane can improve the dispersibility of barite.

To quantify the adsorption amount of waterborne polyurethane (WPU) on the surface of barite particles, thermogravimetric analysis (TG/DTG) was conducted. Figure 7a illustrates the TG/DTG curves of WPU. The TG curve indicates that WPU begins to lose weight starting at 30 °C, primarily due to the high-water content in the prepolymer emulsion. As temperature increases, WPU undergoes continuous weight loss until it approaches zero. The thermal degradation of WPU can be divided into two stages [24]: the decomposition of the hard segment and the soft segment. According to the DTG curve, WPU exhibits three distinct stages of mass loss. The initial stage (216~292 °C) is attributed to the evaporation of bound water within WPU; the second stage (292~371 °C) corresponds to the degradation of the hard segment region, with a peak weight loss temperature at 352.5 °C. The higher internal cohesion and more ordered aggregation of the hard segment result in a higher thermal decomposition temperature [24,25]. The third stage of mass loss (above 371 °C) is associated with the decomposition of the soft-segment region, with a peak weight loss temperature of 432.5 °C. Figure 7b presents the TG/DTG curves of barite before and after modification with WPU. From the TG curve, it is evident that unmodified barite exhibits a weight loss of 1.46%, mainly due to residual impurities in the mineral. After modification, the weight loss of barite is 2.84%, with a difference in weight of 1.38% compared to its original state, indicating that 1.38% of polyurethane has been adsorbed onto the barite. The DTG curve reveals that the weight loss of modified barite occurs in two distinct stages: the first stage from 100 °C to 320 °C and the second stage from 340 °C to 420 °C. This suggests that the hard segment of waterborne polyurethane adsorbed on barite becomes increasingly disordered, resulting in a notable decrease in the decomposition temperature of the hard segment.

Figure 7c illustrates the FT-IR spectra of barite before and after treatment with waterborne polyurethane. In the spectrum of waterborne polyurethane, the peaks at 3312 cm^−1^, 1535 cm^−1^, and 773 cm^−1^ correspond to the stretching vibration of the N-H bond, the in-plane bending vibration, and the out-of-plane bending vibration, respectively. The peaks at 2946 cm^−1^ and 2856 cm^−1^ are attributed to the asymmetric and symmetric stretching vibrations of CH groups [26], while those at 1463 cm^−1^ and 1368 cm^−1^ represent the rocking vibrations of CH_2_ [27]. The peak at 1696 cm^−1^ corresponds to the stretching vibration of C=O. Additionally, the peaks at 1238 cm^−1^ and 1102 cm^−1^ are associated with the stretching vibrations of C-O-C bonds, and the peak at 1304 cm^−1^ is due to the stretching vibration of C-N. In the spectrum of untreated barite, the peaks at 1184 cm^−1^, 1105 cm^−1^, and 1051 cm^−1^ are attributed to the antisymmetric stretching vibrations of ${\mathrm{SO}}_{4}^{2-}$, and the peaks at 632 cm^−1^ and 598 cm^−1^ correspond to the out-of-plane bending vibrations of ${\mathrm{SO}}_{4}^{2-}$ [28].

By comparing the infrared spectra before and after modification, it is evident that barite modified with waterborne polyurethane exhibits three additional absorption peaks in the regions of 1650–1750 cm^−1^ and 2800–3000 cm^−1^. A comparison with the infrared spectrum of waterborne polyurethane reveals precise matches at 1696 cm^−1^ for the C=O stretching vibration peak, 2946 cm^−1^ for the antisymmetric CH_3_ stretching vibration peak, and 2856 cm^−1^ for the symmetric CH_2_ stretching vibration peak. These findings confirm the successful loading of waterborne polyurethane onto the surface of barite.

To further verify the nature of adsorption, X-ray photoelectron spectroscopy (XPS) analysis was conducted on the modified barium sulfate WPU-B, as illustrated in Figure 8. Figure 8a,b present the overall XPS spectra of the samples before and after modification, respectively. A nitrogen peak emerges post-modification, which was not observed in the unmodified sample. Figure 8c,d display the Ba 3d XPS spectra of barium sulfate before and after modification with waterborne polyurethane. The Ba 3d orbital undergoes spin–orbit splitting into 3d3/2 and 3d5/2 suborbitals with distinct binding energies. Pre-modification, the binding energies of the Ba 3d5/2 and Ba 3d3/2 peaks are 795.31 eV and 780.01 eV [29], respectively. After modification, these values shift to 795.30 eV and 780.03 eV, corresponding to shifts of −0.01 eV and +0.02 eV, respectively. Figure 8e,f show the S 2p XPS spectra, where the S 2p orbital also undergoes spin–orbit splitting into 2p3/2 and 2p1/2 components. Before modification, the S 2p3/2 peak was located at 169.67 eV and the S 2p1/2 peak at 168.48 eV [30]. After modification, these peaks shifted slightly to 169.66 eV (S 2p3/2) and 168.44 eV (S 2p1/2), corresponding to shifts of −0.01 eV and −0.04 eV, respectively. The minimal displacement in Ba 3d and S 2p XPS peak positions before and after modification can be attributed to measurement uncertainties rather than significant changes in actual peak positions. However, a notable increase in peak intensity is observed post-modification. Panels (g) and (h) illustrate the C 1s spectra before and after modification. Figure 8g shows C-C (284.80 eV) and C-O (285.96 eV) peaks, likely originating from hydrocarbon contaminants, while Figure 8h displays C-C (284.80 eV), C-O/C-N (286.19 eV), and C=O (289.40 eV) peaks. Compared to panel (g), the post-modification C 1s spectrum exhibits increased peak intensity, a shift in the C-O peak position by 0.27 eV, and the emergence of a new C=O peak at 289.40 eV. These C-O/C-N and C=O peaks are attributed to the -COONH- units in waterborne polyurethane. No N 1s peak was detected pre-modification in panel (i), but post-modification, shown in panel (j), there is a C-N/N-H peak at 399.75 eV, also derived from the -COONH- units in waterborne polyurethane [22,31]. The wet modification process successfully grafted a polyurethane film onto the barite surface, weakening the Ba-O and S-O bonds while enhancing the C-O and C-C bonds and introducing C=O, C-N, and N-H peaks due to the presence of -COONH-.

Based on the aforementioned analysis, a mechanism for waterborne polyurethane (WPU)-modified barium sulfate is proposed, as illustrated in Figure 1. Barium sulfate exhibits an island-like structure characterized by ionic Ba-O and S-O bonds. Due to their relatively low bond energy, Ba-O bonds are prone to breaking, thereby exposing a significant number of positively charged Ba^2+^ ions and negatively charged O^2−^ ions in aqueous solution. During the grafting and heating process in WPU emulsion, isocyanates and polyols undergo condensation reactions to form polyurethane urea groups (-NH-CO-O-). The negatively charged -COO^−^ terminals chemically bond with the positively charged Ba^2+^ ions, forming Ba-COOR complexes. As the reaction progresses, additional isocyanate groups and polyols continue to graft onto the Ba^2+^ ions until a polyurethane film fully coats the surface of barium sulfate, thereby significantly enhancing its dispersion.

### 3.3. Performance Analysis of LDPE/WPU-B Composites

Modified barium sulfate was incorporated into LDPE using mixing, open milling, and compression molding methods, with a WPU-B content of 20%. The tensile and bending properties of the composites were evaluated using a universal testing machine. Figure 9a,b present the tensile strength and tensile modulus of LDPE/WPU-B, while Figure 9c,d show the bending strength and bending modulus. The tensile strength, elongation at break, and tensile modulus of pure LDPE are 13.31 MPa, 363.85%, and 66.19 MPa, respectively. Meanwhile, the bending strength, peak strain, and bending modulus are 9.05 MPa, 9.51%, and 210.10 MPa, respectively. The mechanical performance of the composites improves as the WPU content increases, reaching optimal levels at 1.5 wt% WPU. Compared to LDPE/B without WPU, the tensile strength, elongation at break, and tensile modulus increase by 19.13%, 52.22%, and 37.17%, respectively. Relative to pure LDPE, the tensile strength and modulus rise by 8.33% and 22.31%, while the elongation at break remains almost unchanged. Similarly, the bending strength, peak strain, and modulus improve to 9.71 MPa, 10.05%, and 272.39 MPa, representing increases of 34.48%, 10.01%, and 44.68% compared to LDPE/B without WPU, and 7.29%, 5.68%, and 29.64% compared to pure LDPE. However, further increasing the WPU content results in a decline in both tensile and bending properties of the LDPE/WPU-B composites.

As illustrated in the SEM diagrams of LDPE/B and LDPE/WPU-B (Figure 10), these images provide valuable explanation for the changes in the properties of the composite materials. In Figure 10b, the cross-sectional morphology of LDPE/B reveals agglomerated particles within the field of view, with a distinct interface between the particles and LDPE. The addition of B to LDPE results in decreased tensile properties due to particle agglomeration and incompatibility with organic materials. Conversely, Figure 10c,d show that in LDPE/WPU-B composites, B is more uniformly dispersed with minimal agglomeration. Compared to LDPE/B, there is no clear interface between the matrix material LDPE and the reinforcing material WPU-B, indicating that WPU enhances the dispersibility of B and forms a polyurethane film on its surface, thereby increasing the affinity between B and LDPE and improving the tensile and bending properties of the material. However, when the WPU content exceeds 1.5 wt%, excessive WPU can lead to increased agglomeration of WPU-B, causing stress concentration and negatively impacting the properties of the composite material [32].

The mechanical properties of LDPE/WPU-B composites were investigated with varying WPU-B content, as illustrated in Figure 11. With increasing WPU-B content, both the tensile strength and modulus initially increase, reaching 16.42 MPa and 82.25 MPa, respectively, with a 40% WPU-B content. This represents a 23.37% and 24.26% improvement over pure LDPE. However, the elongation at break decreases with higher WPU-B contents, dropping to 90.33% with a 60% WPU-B content, which is a 75.17% reduction compared to LDPE. The addition of WPU-B enhances material strength but introduces interfacial stress, leading to crack formation upon stretching and thus reducing the elongation at break. In contrast, bending performance initially improves and then declines, peaking with a 40% WPU-B content with values of 12.82 MPa and 332.00 MPa for flexural strength and the modulus, respectively, representing increases of 34.81% and 58.10% compared to LDPE.

### 3.4. Radiation Resistance of LDPE/WPU-B Composites

The radiation shielding performance of LDPE/WPU-B was evaluated at tube voltages of 60 kV, 80 kV, 100 kV, and 120 kV. The thickness of the test samples, obtained by stacking three specimens, was approximately 1.2 cm. The results are summarized as follows: Figure 12a illustrates the density changes of the composite materials under different conditions. The incorporation of WPU slightly decreased the density of the composites, whereas an increase in the amount of WPU-B led to a linear increase in the density of LDPE/WPU-B composites. Figure 12b–f depict the variations in RPE, LAC, MAC, lead equivalent, and HVL for both LDPE/B and LDPE/WPU-B composites at different tube voltages. As shown in Figure 12b–e, with increasing amounts of WPU-B, the evaluation metrics RPE, LAC, MAC, and lead equivalent of the composites progressively increased, while HVL decreased (shown in Figure 12f). In the low-energy range, photoelectric absorption is the dominant photon attenuation mechanism [33]. The increased presence of WPU-B particles enhances the probability of incident photons interacting with these particles, resulting in more frequent collisions and photoelectric effects, thereby reducing the energy of the incident photons and their secondary scattering. With the increase in tube voltage, the RPE, LAC, and MAC of materials with different additive amounts decreased to varying degrees, while the HVL increased correspondingly. This phenomenon can be attributed to two primary factors. Firstly, except for special cases such as the K-edge of barium, the photon absorption cross-section usually decreases with the increase in photon energy [17]; Secondly, high-energy rays can cause the breakage of covalent bonds in the LDPE matrix [34]. As energy increases, the degree of covalent bond breakage intensifies, thereby weakening the synergy between LDPE and the reinforcing body of WPU-B particles, and ultimately leading to a decline in radiation shielding performance. The change in lead equivalent varies with different materials. As tube voltage increases, the lead equivalent of LDPE/WPU-B (40%) and LDPE/WPU-B (60%) initially increases before decreasing. This trend is likely due to a large portion of 120 kV photons falling within the weak absorption region of barium, which reduces the radiation shielding efficiency and lead equivalent at a 120 kV tube voltage. Considering all indicators, an LDPE/WPU-B composite material with a 40% WPU-B content demonstrates superior radiation resistance performance. At a tube voltage of 80 kV, the respective parameters are ρ = 1.93 g·cm^−3^, RPE = 97.90%, LAC = 3.21 cm^−1^, MAC = 1.67 cm^2^·g^−1^, lead equivalent = 0.35 mmPb, and HVL = 0.22 cm.

It is worth noting that with the same additional amount (20 wt%), the various performance indicators of LDPE/WPU-B are significantly higher than those of LDPE/B. The primary reason for this difference lies in the superior dispersion of WPU-modified WPU-B particles within the LDPE matrix, as illustrated in Figure 13. When X-rays penetrate the composite material, the enhanced dispersion of WPU-B particles results in more frequent interactions between photons and WPU-B particles. This increases the probability of the photoelectric effect occurring and reduces the energy of secondary scattered photons, thereby enhancing the material’s shielding effectiveness against X-rays [35].

## 4. Conclusions

The novel modifier WPU was used as a wetting agent, enhancing hydrophobicity and improving the dispersibility of barite. Adding 1.5 wt% WPU yielded optimal mechanical properties in LDPE/WPU-B composites, with significant increases in tensile strength, elongation at break, tensile modulus, flexural strength, and flexural modulus. Even with a 40 wt% WPU-B content, the composite retained excellent tensile and flexural properties. The LDPE/WPU-B composite also showed superior X-ray shielding performance, especially at a 80 kV tube voltage. At 80 kV, the parameters were ρ = 1.93 g·cm^−3^, RPE = 97.90%, LAC = 3.21 cm^−1^, MAC = 1.67 cm^2^·g^−1^, lead equivalent = 0.35 mmPb, and HVL = 0.22 cm. The WPU modifier not only broadens the application scope of barite in the plastic industry but also significantly enhances its radiation protection efficacy, offering great potential for advanced radiation shielding materials.

## Figures and Tables

**Figure 1 polymers-17-00451-f001:**
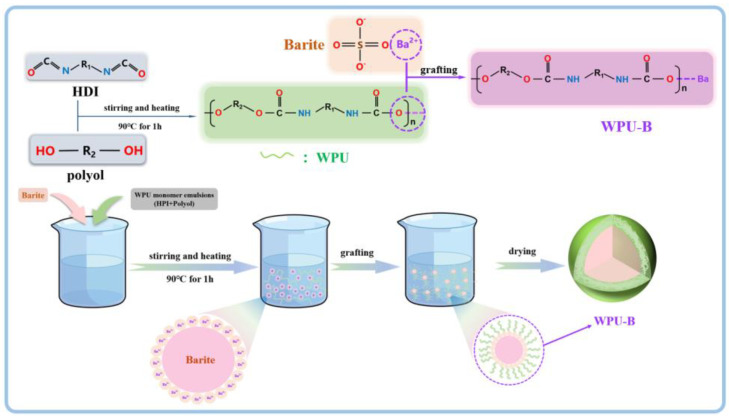
Mechanism of waterborne polyurethane-modified barite.

**Figure 2 polymers-17-00451-f002:**
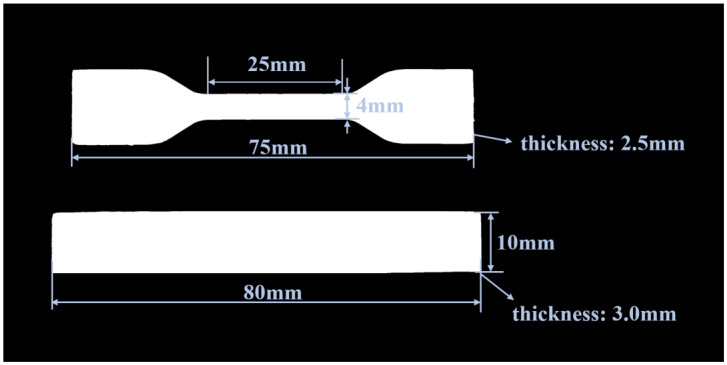
Test specimens of LDPE/WPU-B and LDPE/B composites (dumbbell-type samples are used to test tensile properties, and bar-type samples are used to test bending properties).

**Figure 3 polymers-17-00451-f003:**
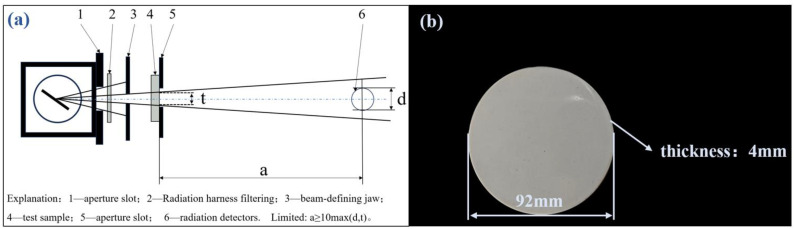
Narrow beam conditions in the radiation shielding test (**a**) and the size of the test sample (**b**).

**Figure 4 polymers-17-00451-f004:**
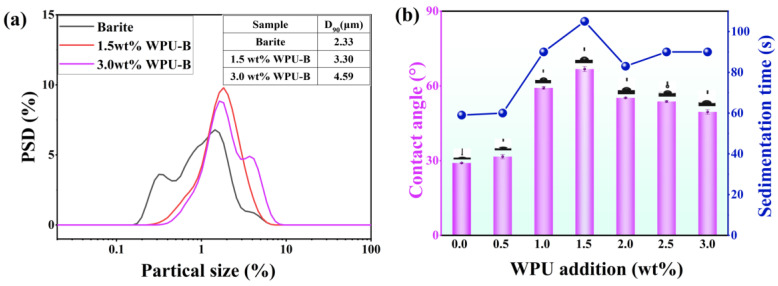
The influence of the additional amount of waterborne polyurethane on the particle size of barite (**a**), sedimentation time, and contact angle (**b**).

**Figure 5 polymers-17-00451-f005:**
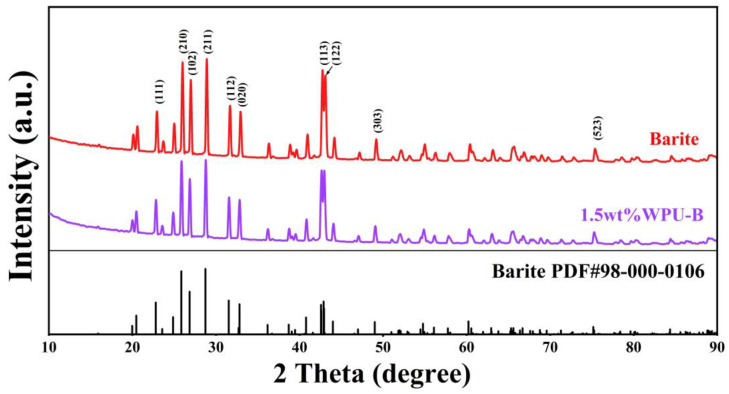
Influence of waterborne polyurethane on XRD of barite.

**Figure 6 polymers-17-00451-f006:**
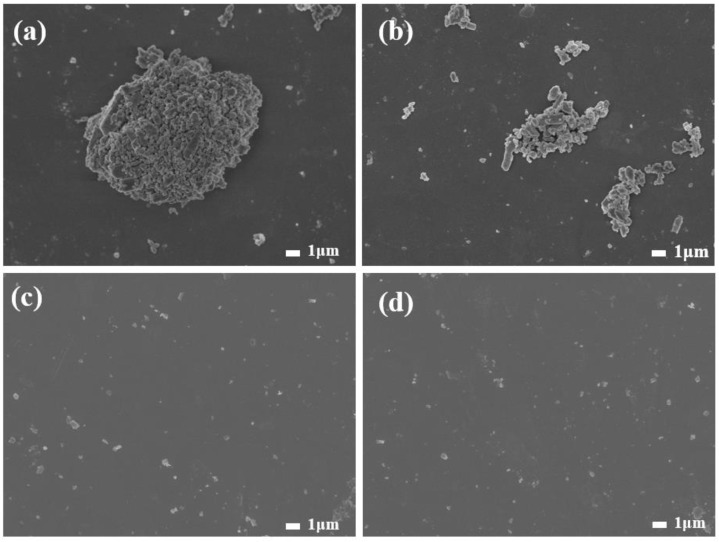
SEM images of barite before (**a**,**b**) and after (**c**,**d**) modification with waterborne polyurethane.

**Figure 7 polymers-17-00451-f007:**
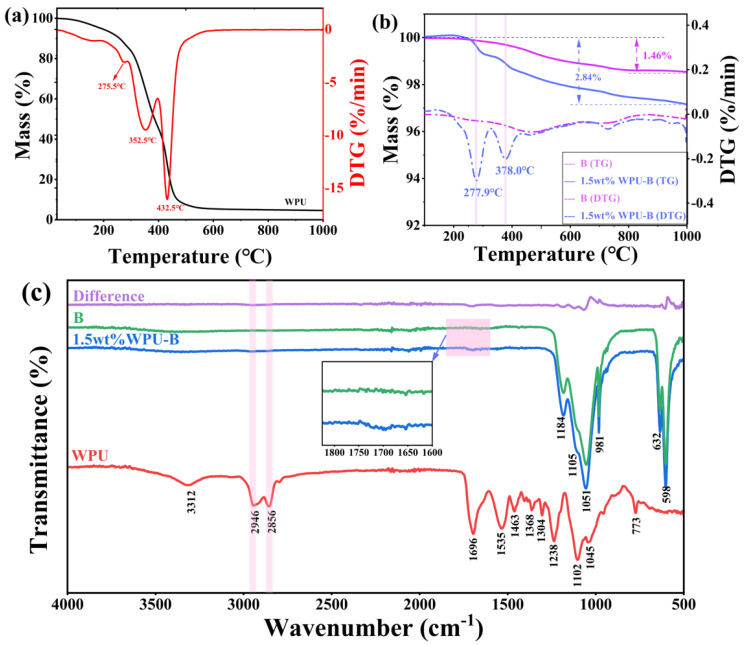
TG/DTG curve of WPU (**a**,**b**) and FT-IR spectrum changes (**c**) of barite before and after modification with waterborne polyurethane.

**Figure 8 polymers-17-00451-f008:**
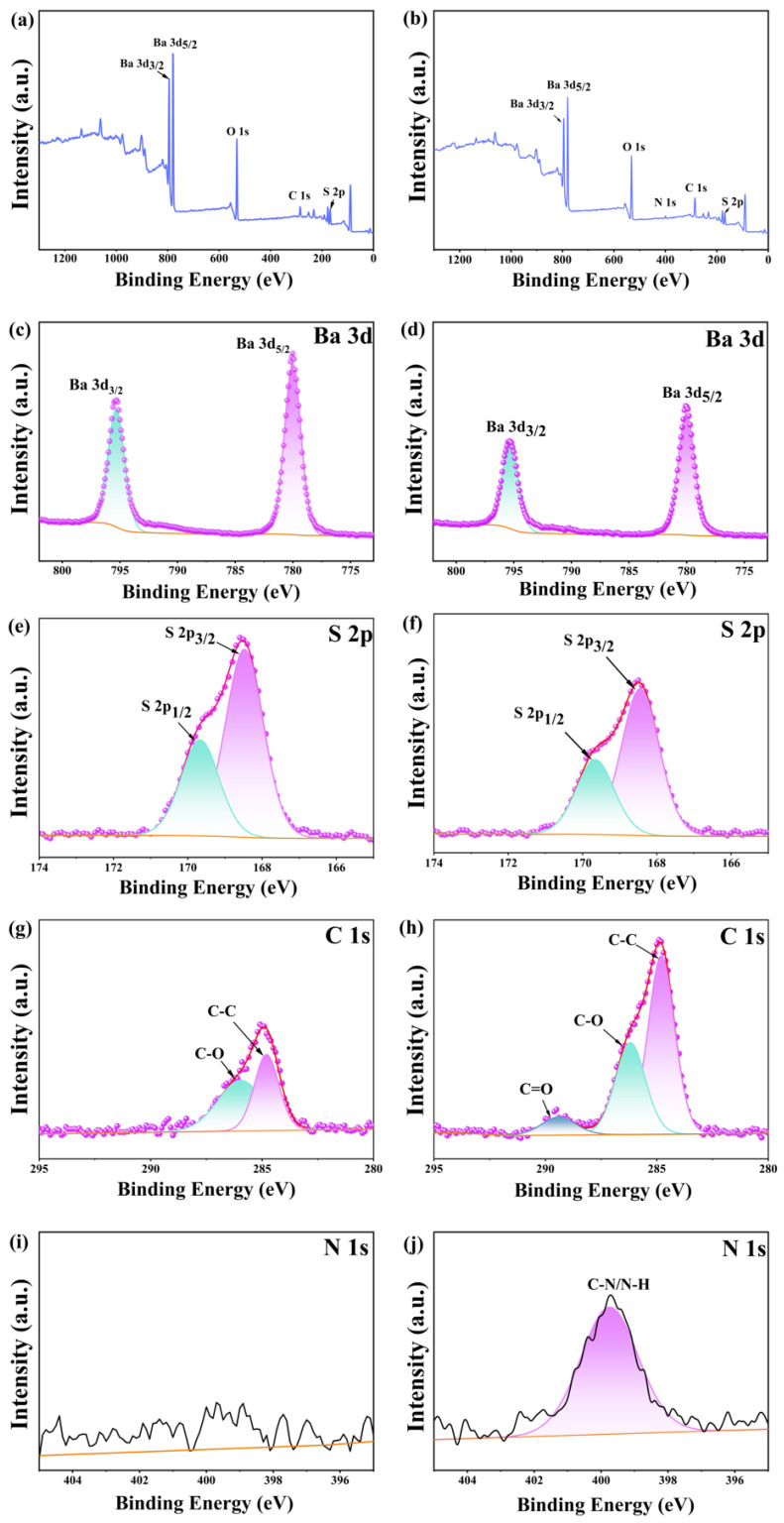
XPS analysis of barite before (**a**,**c**,**e**,**g**,**i**) and (**b**,**d**,**f**,**h**,**j**) after modification with waterborne polyurethane.

**Figure 9 polymers-17-00451-f009:**
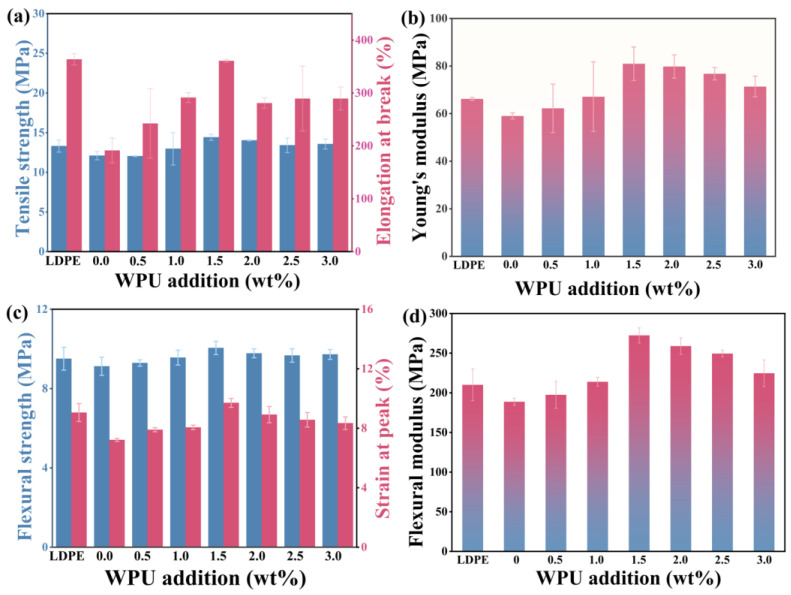
The mechanical property change diagram of LDPE/WPU-B composites with different additional amounts of WPU. (**a**) Tensile strength, (**b**) Young’s modulus, (**c**) Flexural strength, (**d**) Flexural modulus.

**Figure 10 polymers-17-00451-f010:**
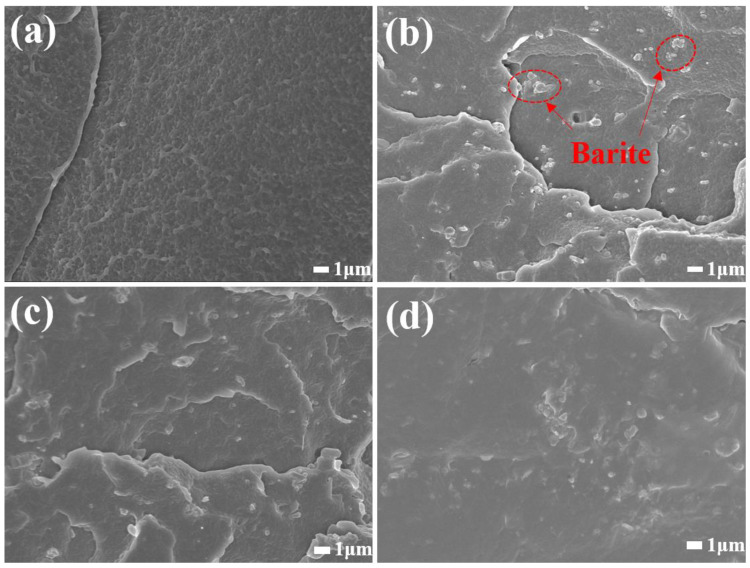
The SEM images of the fracture surfaces of LDPE (**a**), LDPE/B (**b**), and LDPE/WPU-B (**c**,**d**) composites.

**Figure 11 polymers-17-00451-f011:**
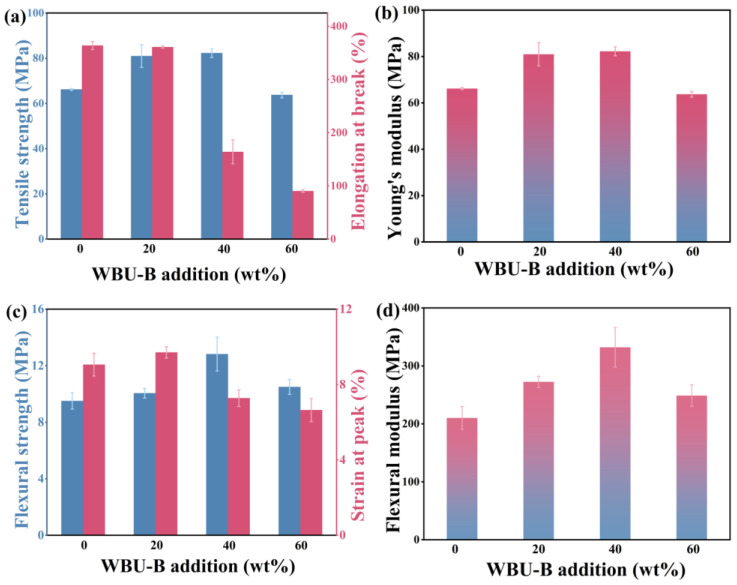
The mechanical property change diagram of LDPE/WPU-B composites with different additional amounts of WPU/B. (**a**) Tensile strength, (**b**) Young’s modulus, (**c**) Flexural strength, (**d**) Flexural modulus.

**Figure 12 polymers-17-00451-f012:**
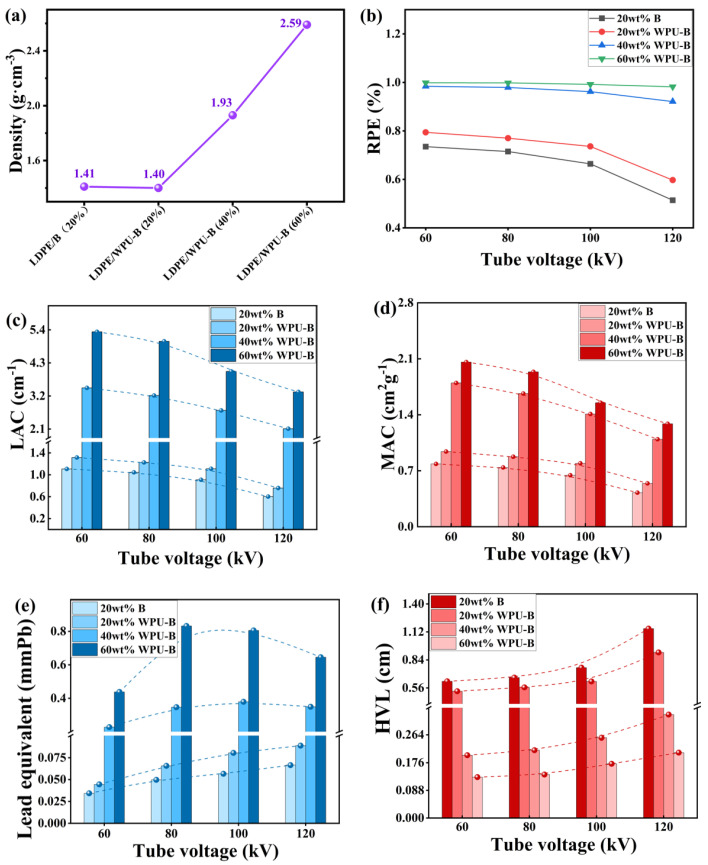
(**a**) The density of the LDPE/WPU-B composite materials, (**b**) the RPE of the LDPE/B and LDPE/WPU-B composite materials, (**c**) the LAC of the LDPE/B and LDPE/WPU-B composite materials, (**d**) the MAC of the LDPE/B and LDPE/WPU-B composite materials, (**e**) the lead equivalent of the LDPE/B and LDPE/WPU-B composite materials, and (**f**) the HVL of the LDPE/B and LDPE/WPU-B composite materials.

**Figure 13 polymers-17-00451-f013:**
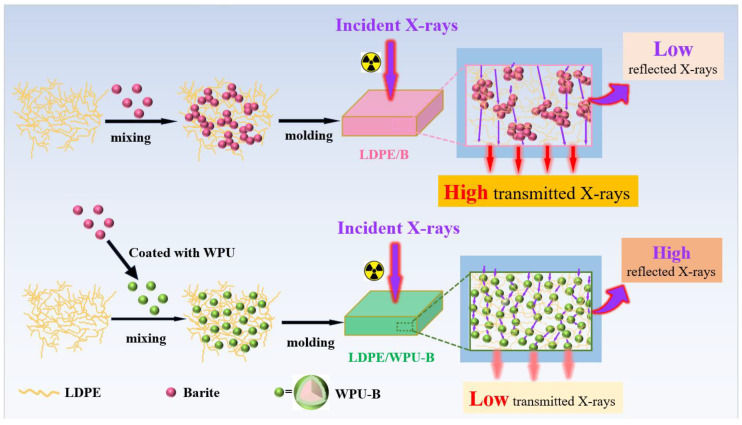
The attenuation mechanism of X-rays in LDPE/B and LDPE/WPU-B composite materials.

**Table 1 polymers-17-00451-t001:** The abbreviations and corresponding explanations.

Abbreviation	Explanation
B	Barite
WPU-B	Barite powder modified with waterborne polyurethane
LDPE	Low-density polyethylene
LDPE/B	Composite material reinforced with barite and matrix composed of low-density polyethylene
LDPE/WPU-B	Composite material reinforced with WPU-B and matrix composed of low-density polyethylene
PSD	Particle size distribution

## Data Availability

The original contributions presented in this study are included in the article. Further inquiries can be directed to the corresponding author.

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
