# Peer review of "Study of Construction of Innovative Barite/Waterborne Polyurethane/Low-Density Polyethylene Composites for Enhanced X-Ray Shielding Performance"

_polymers, 2025, doi:10.3390/polym17040451_

Round 1

Reviewer 1 Report

Comments and Suggestions for Authors

This research lays a solid groundwork for creating safer and more sustainable materials for radiation shielding. 

The topic presented in this paper is relevant. 

This study introduces an innovative WPU-modified barite material that enhances polymer compatibility, significantly improving the mechanical properties and radiation shielding efficiency of composites. 

With 40% WPU-B content, it achieves a lead-equivalent shielding of 0.38 mmPb at 100 kV, providing a practical and non-toxic replacement for lead. It stands out by combining excellent mechanical strength and efficient radiation shielding in a lightweight, integrated solution.

Considering the previously described, the paper may be of interest to the readers of the Polymers journal.

The structure of the paper, divided into the chapters is logical and it is easy to follow and understand the content.

The paper includes extensive list of references. The list of references is appropriate for the topic observed and includes recent scientific studies.

The conclusions are consistent with the evidence and arguments presented and address the main question posed.

Still, in order to improve the content of the paper, some segments of the that need to be improved are as follows:

- The structure of the text within Testing and Analysis methods should be improved (written in full sentences, not like Settling Time: ...).

- Authors should follow the template for the paper preparation. Identifying the highlights of the paper as a special segment, before the introduction is not in accordance with the instructions for authors.

- Figure 2 is not acceptable. The data can be written in a sentence in the text, or the image should be cropped to be presented without the measuring device).

- Figure 4 is not readable.

- Font size on graph axis is too large (fig. 4, 5)

- There are many typing errors, mostly referring to missing spaces between the letters, both in the text and figures.

Author Response

Reviewer #1:

Reviewer Comments and Suggestions for Authors:

This research lays a solid groundwork for creating safer and more sustainable materials for radiation shielding. 

The topic presented in this paper is relevant. 

This study introduces an innovative WPU-modified barite material that enhances polymer compatibility, significantly improving the mechanical properties and radiation shielding efficiency of composites. 

With 40% WPU-B content, it achieves a lead-equivalent shielding of 0.38 mmPb at 100 kV, providing a practical and non-toxic replacement for lead. It stands out by combining excellent mechanical strength and efficient radiation shielding in a lightweight, integrated solution.

Considering the previously described, the paper may be of interest to the readers of the Polymers journal.

The structure of the paper, divided into the chapters is logical and it is easy to follow and understand the content.

The paper includes extensive list of references. The list of references is appropriate for the topic observed and includes recent scientific studies.

The conclusions are consistent with the evidence and arguments presented and address the main question posed.

Still, in order to improve the content of the paper, some segments of the that need to be improved are as follows:

- The structure of the text within Testing and Analysis methods should be improved (written in full sentences, not like Settling Time: ...).

- Authors should follow the template for the paper preparation. Identifying the highlights of the paper as a special segment, before the introduction is not in accordance with the instructions for authors.

- Figure 2 is not acceptable. The data can be written in a sentence in the text, or the image should be cropped to be presented without the measuring device).

- Figure 4 is not readable.

- Font size on graph axis is too large (fig. 4, 5)

- There are many typing errors, mostly referring to missing spaces between the letters, both in the text and figures.

(R1Q1) The structure of the text within Testing and Analysis methods should be improved (written in full sentences, not like Settling Time: ...).

>>AUTHORS’ REPLY:

Thanks for your valuable feedback. We recognize the critical importance of providing a comprehensive and clear description of the texting and analysis method. In response to your suggestions, we have thoroughly revised the relevant sections in the manuscript to ensure that the description of the texting and analysis method are complete, precise, and readily comprehensible.

(R1Q2) Authors should follow the template for the paper preparation. Identifying the highlights of the paper as a special segment, before the introduction is not in accordance with the instructions for authors.

>>AUTHORS’ REPLY:

We sincerely appreciate your meticulous review and valuable feedback. We have reviewed the manuscript and found that this is a formatting issue. We have made the revisions in the revised draft, and the entire text has been proofread and corrected in accordance with the author guidelines.

(R1Q3) Figure 2 is not acceptable. The data can be written in a sentence in the text, or the image should be cropped to be presented without the measuring device).

>>AUTHORS’ REPLY:

Thanks for your valuable feedback. Upon reviewing the manuscript, we have identified that the presentation of Figure 2 is indeed inappropriate. To address this issue, we have removed the schematic diagram of the measuring device and provided a complete and accurate representation in the revised manuscript.

(R1Q4) Figure 4 is not readable.

>>AUTHORS’ REPLY:

We appreciate your valuable suggestions and have revised Figure 4 in the submitted manuscript to ensure it now presents comprehensive information.

(R1Q5) Font size on graph axis is too large (fig. 4, 5).

>>AUTHORS’ REPLY:

Thanks for your valuable suggestions. In response, we have adjusted the font size of the axes in Figures 4 and 5 of the submitted manuscript to ensure that it is both appropriate and easily readable.

(R1Q6) There are many typing errors, mostly referring to missing spaces between the letters, both in the text and figures.

>>AUTHORS’ REPLY:

We sincerely appreciate your thorough review of this manuscript and the valuable feedback provided. In accordance with your suggestions, we have conducted a detailed proofreading and revision process, and the amended contents are now reflected in the resubmitted manuscript.

Reviewer 2 Report

Comments and Suggestions for Authors

As regards the material physics parts of this paper, I can offer no expert opinion, as I have insufficient knowledge and experience in this field, although to my mind there seems to be nothing obvious to object to.  But, in Fig. 4a, I would suggest writing out explicitly what 'PSD' label on the vertical axis stands for — particle size distribution?

As regards the radiological parts of this paper, my comments are as follows.

Lines 153-154.  Give the typical radiation doses that were used when the measurements were made.

Line 391.  Say how the lead equivalent values were determined.  By making measurements with lead test samples as in Fig. 3?

Line 400.  I am a little surprised that the radiation doses likely to have been accumulated in the course of the reported work would have been sufficiently high to lead to cracking of the LDPE matrix.

Lines 401-409.  I do not fully understand what the authors mean here.  Is the explanation of the observations in the sentence in lines 397-399 not simply that photon absorption cross-sections (apart from at edges such as the barium K-edge) generally decrease with increasing photon energy?

Fig. 10b.  Should 'RAR' on the vertical axis be replaced by RPE?

Overall recommendation:  publish after minor revision.

Author Response

(R2Q1) As regards the material physics parts of this paper, I can offer no expert opinion, as I have insufficient knowledge and experience in this field, although to my mind there seems to be nothing obvious to object to.  But, in Fig. 4a, I would suggest writing out explicitly what 'PSD' label on the vertical axis stands for — particle size distribution?

>>AUTHORS’ REPLY:

Thank you for your valuable feedback. After re-examining the manuscript, we found that the use of abbreviations was inappropriate in the text. Specifically, PSD here refers to particle size distribution. To ensure clarity, we have provided a detailed explanation of PSD in the revised version of the manuscript submitted (Line 207).

(R2Q2) Lines 153-154.  Give the typical radiation doses that were used when the measurements were made.

>>AUTHORS’ REPLY:

Thanks for your suggestion. We have included typical radiation dose data in the submitted revised manuscript to ensure the integrity of the data. (Line 172).

(R2Q3) Line 391.  Say how the lead equivalent values were determined.  By making measurements with lead test samples as in Fig. 3?

>>AUTHORS’ REPLY:

Thank you for your valuable suggestions. After re-reviewing the manuscript, we noticed that this was an oversight. The calculation method of lead equivalent involves irradiating the material to be tested with X-rays, measuring its radiation dose, and comparing it to the radiation dose of a lead plate of known thickness under the same conditions, thereby calculating the lead equivalent of the material to be tested. We have presented it clearly in the submitted revised manuscript to ensure the clarity of the data (Line 198~201).

(R2Q4) Line 400.  I am a little surprised that the radiation doses likely to have been accumulated in the course of the reported work would have been sufficiently high to lead to cracking of the LDPE matrix.

>>AUTHORS’ REPLY:

We appreciate your valuable questions and suggestions. Upon reviewing the manuscript, we identified an imprecise expression. Specifically, high-energy incident photons do not directly cause the cracking of the LDPE matrix. Instead, they induce the breakage of covalent bonds within the LDPE, thereby reducing the mechanical properties of the composite material, including tensile and bending strength. This correction has been made in the submitted revised manuscript.

(R2Q5) Lines 401-409.  I do not fully understand what the authors mean here.  Is the explanation of the observations in the sentence in lines 397-399 not simply that photon absorption cross-sections (apart from at edges such as the barium K-edge) generally decrease with increasing photon energy?

>>AUTHORS’ REPLY:

Thanks for your valuable questions and suggestions. Your suggestion concerning the expression in question is indeed appropriate. During the process of re-reviewing the manuscript, we found that some expressions were redundant and not accurate enough. Regarding the reasons for the decrease in the radiation resistance of the material with the increase in tube voltage, they can be summarized as follows: Firstly, except for special cases such as the K-edge of barium, the photon absorption cross-section usually decreases with the increase in photon energy (References: Messele, A. G.; Penev, K. I.; Mequanint, K.; Mekonnen, T. H., Lead-free single and dual-filler loaded polychloroprene X-ray shielding nanocomposites. Applied Materials Today 2025, 42, 102558); Secondly, high-energy rays can cause the breakage of covalent bonds in the LDPE matrix (Bhattacharya, A., Radiation and industrial polymers. Progress in Polymer Science 2000, 25, (3), 371-401). As energy increases, the degree of covalent bond breakage intensifies, thereby weakening the synergy between LDPE and the reinforcing body WPU-B particles, and ultimately leading to a decline in radiation shielding performance (Line 429~435).

(R2Q6) Fig. 10b.  Should 'RAR' on the vertical axis be replaced by RPE?

>>AUTHORS’ REPLY:

Thanks for your meticulous attention to detail. The "RAR" here is a wrong expression. The correct expression should be "RPE". We have made the correction in the submitted revised manuscript.

Round 2

Reviewer 1 Report

Comments and Suggestions for Authors

Accept